# Soft Switching DC Converter for Medium Voltage Applications

**Bor-Ren Lin** 

Department of Electrical Engineering, National Yunlin University of Science and Technology, Yunlin 640, Taiwan; linbr@yuntech.edu.tw; Tel.: +886-912312281

**Abstract:** A dc-dc converter with asymmetric pulse-width modulation is presented for medium voltage applications, such as three-phase ac-dc converters, dc microgrid systems, or dc traction systems. To overcome high voltage stress on primary side and high current rating on secondary side, three dc-dc circuits with primary-series secondary-parallel structure are employed in the proposed converter. Current doubler rectifiers are used on the secondary side to achieve low ripple current on output side. Asymmetric pulse-width modulation is adopted to realize soft switching operation for power switches for wide load current operation and achieve high circuit efficiency. Current balancing cells with magnetic component are used on the primary side to achieve current balance in each circuit cell. The voltage balance capacitors are also adopted on primary side to realize voltage balance of input split capacitors. Finally, the circuit performance is confirmed and verified from the experiments with a 1.44 kW prototype.

**Keywords:** soft switching; asymmetric pulse-width modulation (APWM) converter; current doubler rectifier

## 1. Introduction

Medium voltage dc–dc converters have been proposed and implemented to achieve high power density and high efficiency advantages for dc light rail transportation systems [1,2], dc microgrid systems [3,4], or industry power converters [5,6]. In those applications, the high side dc bus voltage is normally at 750 V. The 1200 V SiC or Insulated Gate Bipolar Transistor (IGBT) power switches can be used to convert 750 V dc bus voltage to low voltage output through high-frequency link dc–dc converters. However, SiC devices are expensive, and the switching frequency of IGBT devices is less than 60kHz. The series-connected switches [7,8] and series-connected dc–dc converters [9,10] can be used for medium voltage converters with 600V Metal-Oxide-Semiconductor Field-Effect Transistors (MOSFETs) power devices. These two approaches can reduce the voltage stress on power devices. However, the voltage stresses on each power switch are difficult and unbalanced. Therefore, power devices still have an unbalanced voltage stress problem. Three-level pulse-width modulation converters or resonant converters have been presented in [11–14] to lessen the voltage rating and switching loss on power devices. Modular converters with series or parallel connection have been developed in [15–17] for high voltage or current applications. However, the current balance in each circuit modular should be controlled well in order to distribute equal power in each modular. To solve the current balance issue in each circuit modular, the current balance control approaches have been discussed in [18,19] by using the passive magnetic component.

This paper presents a high voltage dc–dc converter with three cascade half-bridge circuits on primary side to reduce the voltage and current ratings of power devices for high voltage and medium power applications, such as dc light trail transportation systems and three-phase ac–dc power converters. Voltage balance capacitors are also employed on the primary side in order to balance

input split voltages. To prevent current imbalance on each half bridge circuit, the magnetic coupling (MC) current balance components are employed between each half bridge circuit. If the primary-side currents are unbalanced, then the primary-side and secondary-side voltages of MC component will decreased, or increased, in order to compensate the imbalance in primary-side currents. Asymmetric pulse-width modulation approach is adopted to realize the soft switching turn-on characteristic for power switches. Therefore, the switching losses of power devices at high frequency operation can be reduced. Current doubler rectifiers are used on low voltage side in order to reduce the output ripple current. The paper is organized as follows. The circuit diagram and operating principle are presented in Section 2. The circuit characteristics of the proposed converter are discussed in Section 3. In Section 4, experiments are provided to demonstrate the effectiveness of the developed circuit. Then, the conclusion of the presented circuit is discussed in Section 5.

## 2. Circuit Diagram and Principles of Operation

The developed high-frequency link dc–dc converter is illustrated in Figure 1 to realize the main benefits of soft switching operation, low switching loss, low output ripple current and the balance primary-side and secondary-side currents. Three half-bridge circuits with primary-series secondary-parallel configuration are used in the proposed converter to realize low voltage stress $V_{in}/3$ on power switches $S_1 \sim S_6$, and low current stress $I_o/6$ on power diodes $D_1 \sim D_6$. Therefore, 600 V power MOSFETs are used on the primary side to achieve high frequency operation and low conduction loss. Voltage balance capacitors $C_{f1}$ and $C_{f2}$ are employed on the primary side to achieve a split voltages balance at $V_{in}/3$. Half-bridge circuits are operated under asymmetric pulse-width modulation. Therefore, the soft switching turn-on of power switches can be achieved, and the circuit efficiency is improved. The magnetic coupling current balance [19] cells, $MC_1$ and $MC_2$, are used on the primary side to achieve current balance for each half-bridge circuit. Therefore, the current unbalance issue of modular converters is overcome. The current doubler rectifiers are used on the secondary side to accomplish low ripple current on load side.

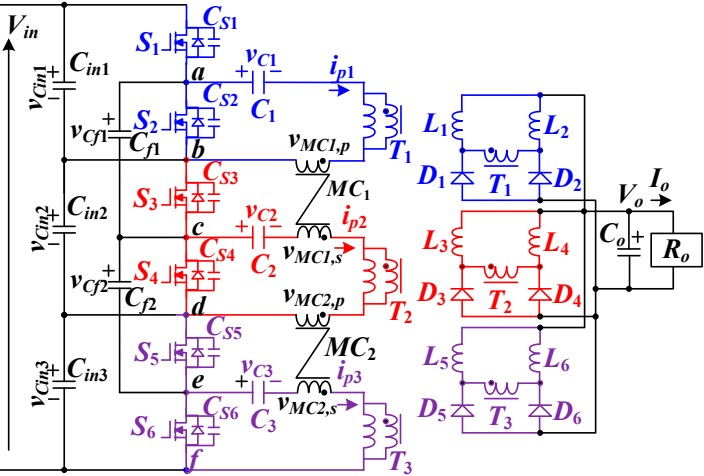

**Figure 1.** Circuit configuration of the developed modular converter for medium voltage applications.

Figure 2 provides the voltage and current waveforms of the studied converter in a switching cycle. Based on the pulse-width modulation waveforms shown in Figure 2, eight operating steps can be observed in each switching cycle under steady state. Power switches $S_1$, $S_3$, and $S_5$ have the same gating signals, and $S_2$, $S_4$, and $S_6$ have the same gating signals. The duty cycle of $S_1$, $S_3$, and $S_5$ denotes $d$ and the duty cycle of $S_2$, $S_4$, and $S_6$ is $1 - d$. When $S_1$, $S_3$, and $S_5$ are active, then $S_2$, $S_4$, and $S_6$ are inactive. We can obtain that $v_{Cf1} = v_{Cin1}$ and $v_{Cf2} = v_{Cin2}$. If $S_1$, $S_3$, and $S_5$ are inactive, and $S_2$, $S_4$, and $S_6$ are active, then $v_{Cf1} = v_{Cin2}$ and $v_{Cf2} = v_{Cin3}$. For steady state operation, the capacitor voltages $v_{Cin1} = v_{Cin2} = v_{Cin3} = v_{Cf1} = v_{Cf2} = V_{in}/3$. Before the system analysis, the circuit parameters on

the proposed circuit are assumed as follows: (1) the same voltage balance capacitances $C_{f1} = C_{f2} = C_f$, (2) the same input split capacitances $C_{in1} = C_{in2} = C_{in3} = C_{in}$, (3) the same output capacitances of power switches $C_{S1} = \ldots = C_{S6} = C_{oss}$, (4) the same dc block capacitances $C_1 = C_2 = C_3 = C_c$, (5) the same turns ratio $n_1 = n_2 = n_3 = n$, (6) the identical magnetizing inductances $L_{m1} = L_{m2} = L_{m3} = L_m$, (7) the same leakage inductances $L_{lk1} = L_{lk2} = L_{lk3} = L_{lk} << L_m$, (8) the same output filter inductances $L_1 = \ldots = L_6 = L_o$, and (9) the primary-side and secondary-side voltages of the magnetic coupling (MC) current balance transformers are zero under steady state. The equivalent circuits for eight operating steps are illustrated in Figure 3, and discussed as follows.

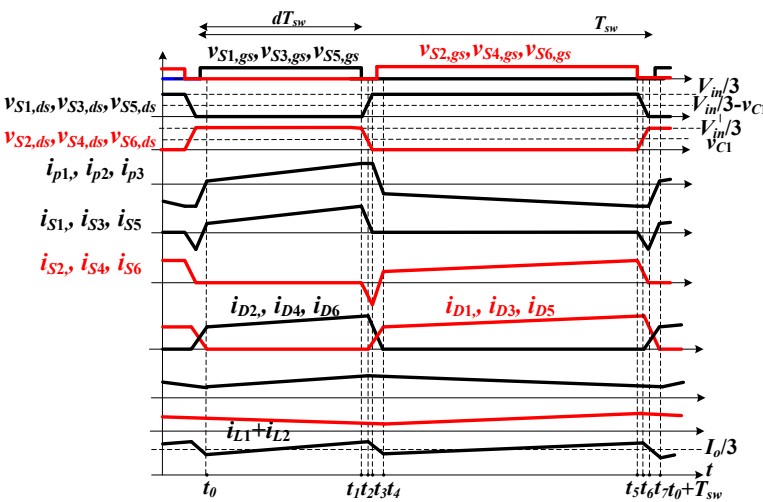

**Figure 2.** Pulse-width modulation waveforms of the proposed converter.

**Step 1 [$t_0 \sim t_1$]:** Before time $t_0$, all the secondary-side diodes are in the commutated interval, and the primary-side currents $i_{p1} \sim i_{p3}$ increase. After time $t_0$, the secondary-side diodes, $D_1$, $D_3$, and $D_5$, are off. In step 1, the primary-side capacitor voltage $v_{Cf1}$ equals $v_{Cin1}$, and $v_{Cf2}$ equals $v_{Cin2}$. The ac-side input voltages of three half bridge circuits $v_{ab}$, $v_{cd}$, and $v_{ef}$ are clamped at $v_{Cin1}$, $v_{Cin2}$, and $v_{Cin3}$, respectively. The magnetizing inductor voltages $v_{Lm1}$, $v_{Lm2}$, and $v_{Lm3}$ are equal to $V_{in}/3 - v_{C1}$, $V_{in}/3 - v_{C2}$, and $V_{in}/3 - v_{C3}$, respectively. The secondary-side inductor voltages $v_{L1}$, $v_{L3}$, and $v_{L5}$ are equal to $(V_{in}/3 - v_{C1})/n - V_o$, $(V_{in}/3 - v_{C2})/n - V_o$ and $(V_{in}/3 - v_{C3})/n - V_o$, respectively, and $v_{L2} = v_{L4} = v_{L6} = -V_o$. Therefore, the primary-side currents $i_{p1} \sim i_{p3}$ and the secondary-side currents $i_{L1}$, $i_{L3}$, and $i_{L5}$ increase, and the output inductor currents $i_{L2}$, $i_{L4}$, and $i_{L6}$ decrease, in step 1.

**Step 2 [$t_1 \sim t_2$]:** Switches $S_1$, $S_3$, and $S_5$ are turned off at time $t_1$. Due to the positive value of $i_{p1} \sim i_{p3}$, $v_{CS1}$, $v_{CS3}$, and $v_{CS5}$ are increased and $v_{CS2}$, $v_{CS4}$, and $v_{CS6}$ are decreased. The charged and discharged times of $C_{S1} \sim C_{S6}$ are very fast so that the primary-side and secondary-side currents are constant in step 2.

**Step 3 [$t_2 \sim t_3$]:** When $v_{CS2} = v_{C1}$, $v_{CS4} = v_{C2}$ and $v_{CS6} = v_{C3}$ at time $t_2$. The primary-side and secondary-side winding voltages of $T_1 \sim T_3$ are zero voltage. Thus, the secondary-side diodes $D_1 \sim D_6$ are forward biased to commutate the load current such that $i_{L1} \sim i_{L6}$ decrease, $i_{D1}$, $i_{D3}$, and $i_{D5}$ increase, and $i_{D2}$, $i_{D4}$, and $i_{D6}$ decrease in step 3. $C_{S2}$, $C_{S4}$, and $C_{S6}$ can be discharged to zero voltage if the energy on the leakage inductors $L_{lk1} \sim L_{lk3}$ is large enough, and the dead time $t_d$ between the upper and lower switches on each half bridge leg is larger than the time interval in steps 2 and 3.

**Step 4 [$t_3 \sim t_4$]:** The capacitor voltages $v_{CS2} = v_{CS4} = v_{CS6} = 0$ at time $t_3$. Due to $i_{p1}(t_3) \sim i_{p3}(t_3)$ being positive, the body diodes of $S_2$, $S_4$, and $S_6$ are conducting so that $S_2$, $S_4$, and $S_6$ are turned on at zero voltage switching. The secondary-side diodes, $D_1 \sim D_6$, are at the commutated interval, and the primary-side leakage inductor voltages are $v_{lk1} = -v_{C1}$, $v_{lk2} = -v_{C2}$, and $v_{lk3} = -v_{C3}$. Thus, the primary-side currents $i_{p1} \sim i_{p3}$ and the secondary-side inductor currents $i_{L1} \sim i_{L6}$ all decrease in step 4.

In step 4, the current variations on $i_{p1} \sim i_{p3}$ approximate $I_o/(3n)$ in order to accomplish the commutation interval through diodes $D_1 \sim D_6$.

Therefore, the duty loss in step 4 is calculated as

$$d_{loss,4} = \frac{\Delta t_{34}}{T_{sw}} \approx \frac{L_{lk}I_o f_{sw}}{3nv_{C1}} \tag{1}$$

**Step 5 [$t_4 \sim t_5$]:** The secondary-side diode currents $i_{D2}$, $i_{D4}$, and $i_{D6}$ are zero, and become reverse biased. In step 5, the capacitor voltages $v_{Cf1} = v_{Cin2}$ and $v_{Cf2} = v_{Cin3}$, and the ac side voltages $v_{ab} = v_{cd} = v_{ef} = 0$. Therefore, the magnetizing inductor voltages $v_{Lm1} \approx -v_{C1}$, $v_{Lm2} \approx -v_{C2}$ and $v_{Lm3} \approx -v_{C3}$. The output inductor voltages $v_{L1} = v_{L3} = v_{L5} = -V_o$, $v_{L2} \approx v_{C1}/n - V_o$, $v_{L4} \approx v_{C2}/n - V_o$ and $v_{L6} \approx v_{C3}/n - V_o$. Thus, $i_{L1}$, $i_{L3}$, and $i_{L5}$ decrease, and $i_{L2}$, $i_{L4}$, and $i_{L6}$ increase, in step 5.

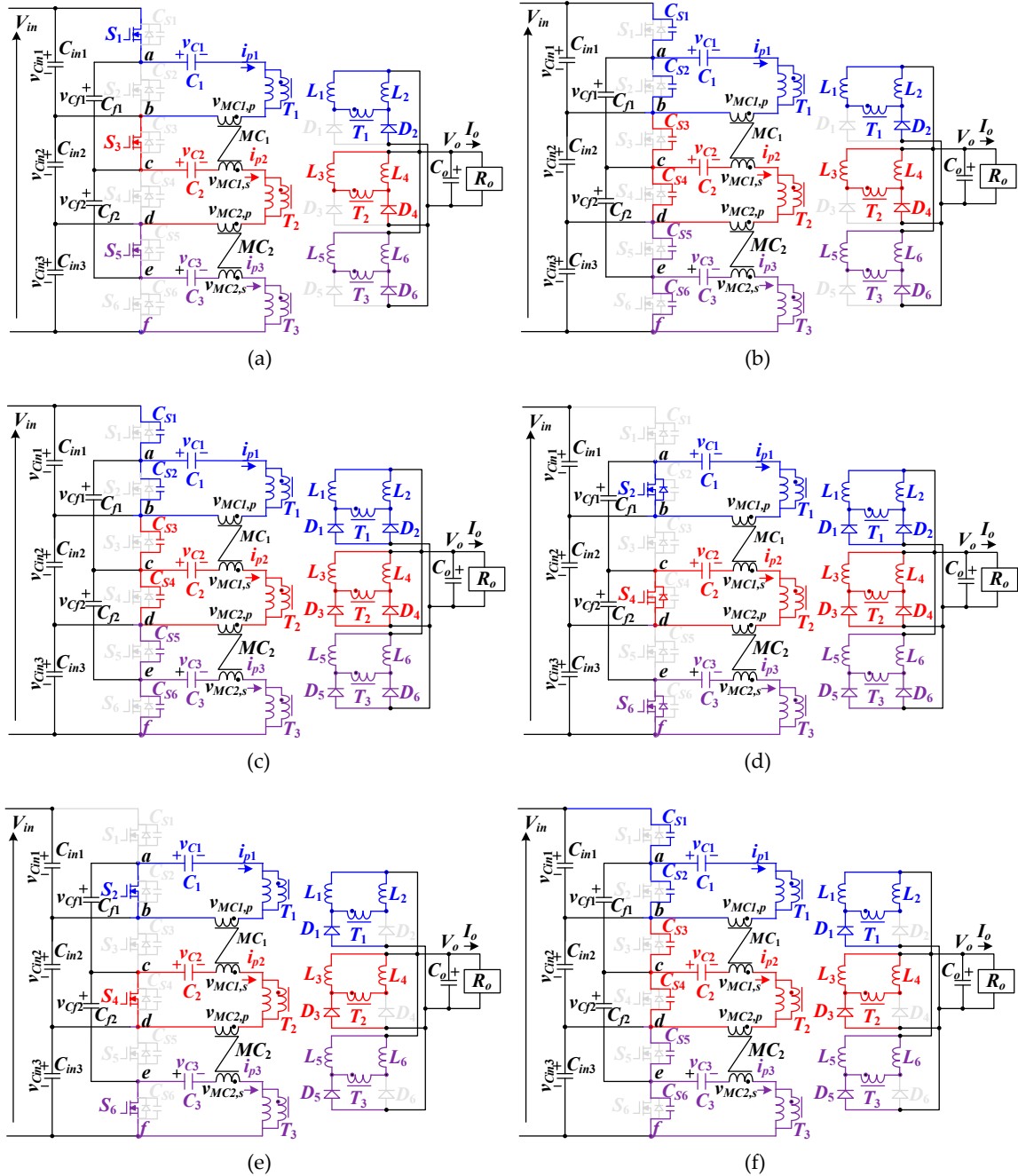

**Figure 3.** *Cont.*

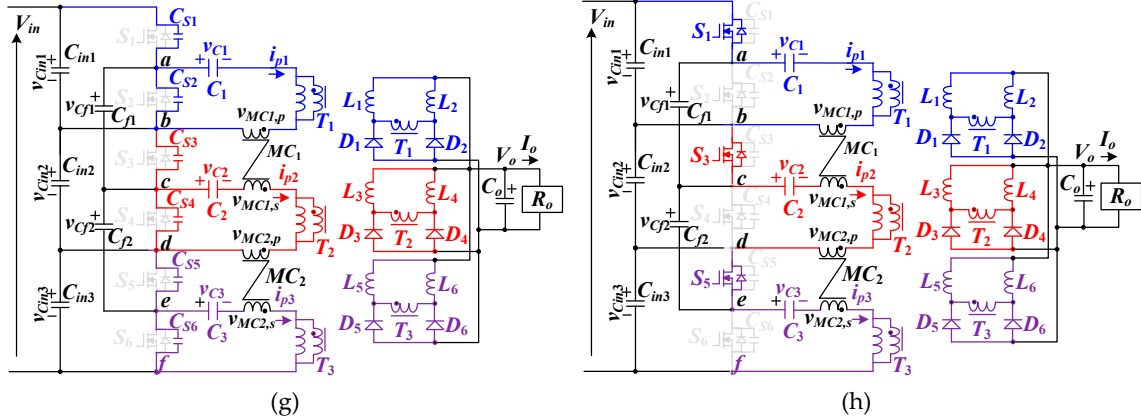

**Figure 3.** Operation steps in a switching period (**a**) step 1, (**b**) step 2, (**c**) step 3, (**d**) step 4, (**e**) step 5, (**f**) step 6, (**g**) step 7, and (**h**) step 8.

**Step 6 [$t_5 \sim t_6$]:** Power switches $S_2$, $S_4$, and $S_6$ are turned off at time $t_5$. Due to $i_{p1}(t_5) < 0$, $i_{p2}(t_5) < 0$ and $i_{p3}(t_5) < 0$, then $C_{S1}$, $C_{S3}$, and $C_{S5}$ will be discharged in step 6. Since the charged and discharged times of $C_{S1} \sim C_{S6}$ are very fast, the primary-side and secondary-side inductor currents are approximately constant in step 6.

**Step 7 [$t_6 \sim t_7$]:** The output capacitor voltages $v_{CS1}$, $v_{CS3}$, and $v_{CS5}$ are discharged to $v_{C1}$, $v_{C2}$, and $v_{C3}$, respectively, at time $t_6$. Then, the primary-side and secondary-side winding voltages of $T_1 \sim T_3$ are all zero voltage. In step 7, the secondary-side diodes $D_1 \sim D_6$ are all conducting to commutate the inductor currents $i_{L1} \sim i_{L6}$. At time $t_7$, $C_{S1}$, $C_{S3}$, and $C_{S5}$ are discharged to zero voltage.

**Step 8 [$t_7 \sim t_0 + T_{sw}$]:** The capacitor voltages $C_{S1}$, $C_{S3}$, and $C_{S5}$ are discharged to zero voltage at time $t_7$. Due to $i_{p1}(t_7) \sim i_{p3}(t_7)$ being all negative, the body diodes of $S_1$, $S_3$, and $S_5$ conduct. Therefore, $S_1$, $S_3$, and $S_5$ can be turned on under zero voltage after time $t_7$. Since $D_1 \sim D_6$ are still conducting, the primary-side currents increase. The diodes currents $i_{D1}$, $i_{D3}$, and $i_{D5}$ decrease to zero at time $t_0 + T_{sw}$. The duty loss in this step 8 is calculated as:

$$d_{loss,8} \approx \frac{L_{lk} I_o f_{sw}}{3n(V_{in}/3 - v_{C1})}. \tag{2}$$

## 3. Circuit Characteristics

Three half bridge circuits are used in the proposed converter with primary-series and secondary-parallel connection to distribute load power through three circuits. Therefore, the power rating of each half bridge circuit is one-third of load power, $P_o/3$. In order to balance the modular currents, the magnetic coupling (MC) current balance cells are presented and discussed in [19]. Therefore, the magnetic coupling current balance components $MC_1$ and $MC_2$ are used in the proposed converter to achieve current balance issue between three half bridge circuits. If the primary-side currents are unbalanced, such as $|i_{p2}| > |i_{p3}|$, then the induced voltage $v_{MC2,p}$ is lessened to decrease current $i_{p2}$ and the secondary-side induced voltage $v_{MC2,s}$ is increased to rise current $i_{p3}$. If the primary currents are balanced ($|i_{p1}| = |i_{p2}| = |i_{p3}|$), then the induced primary-side and secondary-side voltages of $MC_1$ and $MC_2$ cells are zero voltage, $v_{MC1,p} = v_{MC1,s} = v_{MC2,p} = v_{MC2,s} = 0$. Current doubler rectifiers are adopted on the secondary side to lessen the output ripple current. The average dc blocking voltages $V_{C1} \sim V_{C3}$ are related to the input voltage and duty cycle of $S_1$, $S_3$, and $S_5$. These three voltages can be calculated from the flux balance on the primary-side inductors.

$$V_{C1} = V_{C2} = V_{C3} = dV_{in}/3 \tag{3}$$

The output voltage is related to input voltage, duty cycle, and turns ratio, according to the flux balance on the output inductors.

$$V_o = \frac{d(1-d)V_{in}}{3n} - \frac{I_o L_r f_{sw}}{3n^2} - V_f, \tag{4}$$

where $V_f$ is the voltage drop on $D_1 \sim D_6$. Due to the MC, current balance components are used on the primary side of the proposed converter, and the output currents of three half bridge circuits are balanced in steady state. The average secondary-side winding currents of $T_1 \sim T_3$ are zero. Therefore, the average output inductor currents are calculated as

$$I_{L1} = I_{L3} = I_{L5} = (1-d)I_o/3, \ I_{L2} = I_{L4} = I_{L6} = dI_o/3. \tag{5}$$

If the duty cycle $d < 0.5$, then the average inductor currents $I_{L1}$, $I_{L3}$, and $I_{L5}$ are greater than $I_{L2}$, $I_{L4}$, and $I_{L6}$. The ripple currents on $L_1 \sim L_6$ are calculated as

$$\Delta i_{L1} = \Delta i_{L3} = \Delta i_{L5} = \frac{V_o(1-d)T_{sw} + \dfrac{V_o L_{lk} I_o}{n(1-d)V_{in}}}{L_o}, \tag{6}$$

$$\Delta i_{L2} = \Delta i_{L4} = \Delta i_{L6} = \frac{V_o d T_{sw} + \dfrac{V_o L_{lk} I_o}{n d V_{in}}}{L_o}. \tag{7}$$

From (5)~(7), the maximum and minimum output inductor currents are derived as

$$i_{L1,max} = i_{L3,max} = i_{L5,max} = \frac{(1-d)I_o}{3} + \frac{V_o(1-d)T_{sw} + \dfrac{V_o L_{lk} I_o}{n(1-d)V_{in}}}{2L_o}, \tag{8}$$

$$i_{L1,min} = i_{L3,min} = i_{L5,min} = \frac{(1-d)I_o}{3} - \frac{V_o(1-d)T_{sw} + \dfrac{V_o L_{lk} I_o}{n(1-d)V_{in}}}{2L_o}, \tag{9}$$

$$i_{L2,max} = i_{L4,max} = i_{L6,max} = \frac{dI_o}{3} + \frac{d V_o T_{sw} + \dfrac{V_o L_{lk} I_o}{n d V_{in}}}{2L_o}, \tag{10}$$

$$i_{L2,min} = i_{L4,min} = i_{L6,min} = \frac{dI_o}{3} - \frac{d V_o T_{sw} + \dfrac{V_o L_{lk} I_o}{n d V_{in}}}{2L_o}, \tag{11}$$

If the magnetizing inductances of $T_1 \sim T_3$ are given, the ripple currents on $L_{m1} \sim L_{m3}$ are calculated as

$$\Delta i_{Lm} \approx \frac{d(1-d)V_{in}T_{sw} - \dfrac{L_{lk}I_o}{n}}{3L_m}. \tag{12}$$

Due to the conducting time on $D_1 \sim D_6$ being related to the duty cycle $d$, the secondary-side diode average currents and voltage ratings are calculated as

$$I_{D1} = I_{D3} = I_{D5} = (1-d)I_o/3, \ I_{D2} = I_{D4} = I_{D6} = dI_o/3, \tag{13}$$

$$V_{D1,stress} = V_{D3,stress} = V_{D5,stress} = (1-d)V_{in}/(3n), \ V_{D2,stress} = V_{D4,stress} = V_{D6,stress} = dV_{in}/(3n). \tag{14}$$

Since the balance capacitors $C_{f1}$ and $C_{f2}$ are used on the primary side of three half bridge circuits, the input split voltages $V_{Cin1}$, $V_{Cin2}$, and $V_{Cin3}$ are balanced at $V_{in}/3$. Based on the half bridge circuit topology, the voltage rating of power switches $S_1 \sim S_6$ is clamped at $V_{in}/3$. If the ripple

currents on the output inductors and the magnetizing inductors are much less than the average output inductor currents at full load, then the root mean square (*rms*) currents of power devices are expressed in (15) and (16).

$$i_{S1,rms} = i_{S3,rms} = i_{S5,rms} \approx \frac{(1-d)I_o}{3n}\sqrt{d}, \tag{15}$$

$$i_{S2,rms} = i_{S4,rms} = i_{S6,rms} \approx \frac{dI_o}{3n}\sqrt{1-d}. \tag{16}$$

The zero voltage conditions of power switches $S_1$, $S_3$, and $S_5$ are related to the primary-side currents $i_{p1}(t_5)$, $i_{p2}(t_5)$, and $i_{p3}(t_5)$, respectively, and the input voltage. Similar, the zero voltage conditions of power switches $S_2$, $S_4$, and $S_6$ are related to the primary-side currents $i_{p1}(t_1)$, $i_{p2}(t_1)$, and $i_{p3}(t_1)$, respectively, and the input voltage. The primary-side currents $i_{p1}\sim i_{p3}$ at time $t_1$ and $t_5$ are related to the load current and the ripple currents on the magnetizing inductors $L_{m1}\sim L_{m3}$, and output inductors $L_1\sim L_6$.

$$i_{p1}(t_1) = i_{p2}(t_1) = i_{p3}(t_1) \approx i_{Lm1,max} + \frac{i_{L1,max}}{n} \approx \frac{d(1-d)V_{in}T_{sw} - \frac{L_{lk}I_o}{n}}{6L_m} + \frac{(1-d)I_o}{3n} + \frac{(1-d)V_oT_{sw} + \frac{V_oL_{lk}I_o}{n(1-d)V_{in}}}{2nL_o}, \tag{17}$$

$$i_{p1}(t_5) = i_{p2}(t_5) = i_{p3}(t_5) \approx i_{Lm2,min} - \frac{i_{L2,max}}{n} \approx -\frac{d(1-d)V_{in}T_{sw} - \frac{L_{lk}I_o}{n}}{6L_m} - \frac{dI_o}{3n} - \frac{dV_oT_{sw} + \frac{V_oL_{lk}I_o}{ndV_{in}}}{2nL_o}. \tag{18}$$

The necessary leakage inductance to achieve zero-voltage switching (ZVS) condition of $S_1$, $S_3$, and $S_5$ is expressed in (19).

$$L_{lk} \geq \frac{2C_{oss}(V_{in}/3)^2}{i_{p1}^2(t_5)} = \frac{2C_{oss}V_{in}^2}{9i_{p1}^2(t_5)}. \tag{19}$$

Similar, the necessary leakage inductance to achieve ZVS condition of $S_2$, $S_4$, and $S_6$ is expressed in (20).

$$L_{lk} \geq \frac{2C_{oss}(V_{in}/3)^2}{i_{p1}^2(t_1)} = \frac{2C_{oss}V_{in}^2}{9i_{p1}^2(t_1)} \tag{20}$$

Based on the zero-voltage condition in (19) and (20), the necessary leakage inductance can be calculated in (21).

$$L_{lk} \geq \max\left\{\frac{2C_{oss}V_{in}^2}{9i_{p1}^2(t_1)}, \frac{2C_{oss}V_{in}^2}{9i_{p1}^2(t_5)}\right\}. \tag{21}$$

If the ripple voltage on dc blocking capacitors $C_1\sim C_3$ is less than 20% of the average voltage value, then the dc blocking capacitances $C_1\sim C_3$ are calculated as

$$C_1 = C_2 = C_3 > \frac{5(1-d)I_oT_{sw}}{nV_{in}}. \tag{22}$$

## 4. Experimental Results

Experiments based on a laboratory prototype are provided to verify the effectiveness of the developed circuit. The input dc voltage $V_{in,min}$ = 750 V and $V_{in,max}$ = 800 V, the output voltage $V_o$ = 24 V and the maximum load current $I_o$ = 60 A. The switching frequency $f_{sw}$ = 100 kHz. The circuit components of the laboratory prototype are summary in Table 1. Figure 4 illustrates the picture of the laboratory prototype circuit. Figure 5 presents the test results of the gating signals of $S_1$, $S_2$, $S_3$, and $S_5$ under the rated power. The switches $S_1$ and $S_2$ have complementary gating signals, and $S_1$, $S_3$, and $S_5$ have identical pulse-width modulation signals. Figure 6 illustrates the test waveforms of the primary-side voltage $v_{ab}$ and currents $i_{p1}\sim i_{p3}$ at the rated power. It is clear that the three primary-side currents $i_{p1}\sim i_{p3}$ are well balanced. The input split voltages and balance capacitor voltage

at the rated power are measured and presented in Figure 7. The measured voltages are $V_{Cin1}$ = 251 V, $V_{Cin2}$ = 249.3 V, $V_{Cin3}$ = 249.7 V, $V_{Cf1}$ = 251 V, and $V_{Cf2}$ = 249 V under $V_{in}$ = 750 V input. It is clear that the input split voltages and balance capacitor voltages are well balanced. Figure 8 presents the test results of the gating voltage and drain current of power switch $S_1$ under 20% output load, and the rated output load for both 750 V and 800 V input cases. Before the power switch $S_1$ is turned on, the drain current is a negative value, to discharge the drain voltage. Therefore, the soft switching characteristic of $S_1$ can be realized from 20% output load to the rated output load, based on the test results in Figure 8. Similarly, the measured waveforms of $S_2$ are presented in Figure 9. It can be observed that the zero-voltage switching characteristic of $S_2$ is also achieved from 20% output load. Since power devices $S_3$ and $S_5$ are operated in the same circuit manner as power device $S_1$, and $S_4$ and $S_6$ are controlled in the same circuit manner as $S_2$, it can be expected that the zero-voltage switching of power switches $S_3$~$S_6$ are all achieved from 20% output load. The secondary-side inductor currents and diode currents of first half bridge circuit are measured and presented in Figure 10a under the rated power. The output currents of three half bridge circuits under the full output power are illustrated in Figure 10b. The test results show the three output currents are well balanced. Figure 11 illustrates the measured efficiencies of the proposed circuit under different load conditions and input voltages. The measured circuit efficiencies of the developed converter are 91.5% at 20% rated power, 93.2% at 50% rated power, and 92.7% at the rated power under 750 V input. At 750 V input and the rated power, the duty ratio is close to 0.5, and the current rating on power devices and inductors are almost balanced. Therefore, the power losses are nearly distributed into each power devices and the circuit efficiency is improved, compared to the low load condition and higher voltage input.

**Table 1.** Prototype Circuit Parameters.

| Items | Symbol | Parameter |
|---|---|---|
| Input voltage | $V_{in}$ | 750 V~800 V |
| Output voltage | $V_o$ | 24 V |
| Rated output current | $I_o$ | 60 A |
| Switching frequency | $f_{sw}$ | 100 kHz |
| Input capacitors | $C_{in1}, C_{in2}, C_{in3}$ | 180 µF/450 V |
| Voltage balance capacitors | $C_{f1}, C_{f2}$ | 2.2 µF/630 V |
| Power switches | $S_1$~$S_6$ | 2SK4124 |
| Rectifier diodes | $D_1$~$D_6$ | MBR40100PT |
| dc block capacitors | $C_1$~$C_3$ | 0.2 µF |
| Turns ratio of $T_1$~$T_3$ | $n_1$~$n_3$ | 1.5 (33 turns/22 turns) |
| Leakage inductances | $L_{lk1}$~$L_{r3}$ | 15 µH |
| Magnetizing inductances | $L_{m1}$~$L_{m3}$ | 0.6 mH |
| Output inductances | $L_1$~$L_6$ | 66 µH |
| Output capacitance | $C_o$ | 4700 µF/50 V |

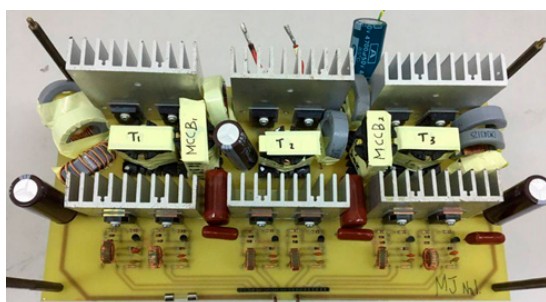

**Figure 4.** Picture of the prototype circuit.

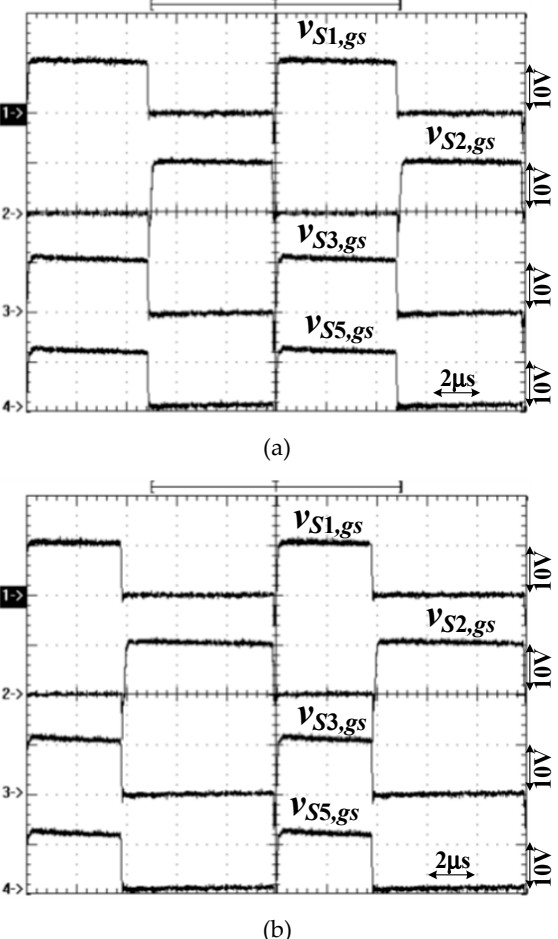

(a)

(b)

**Figure 5.** Measured waveforms of the gating voltages of $S_1$, $S_2$, $S_3$, and $S_5$ at the rated power (**a**) with $V_{in}$ = 750 V and (**b**) with $V_{in}$ = 800 V.

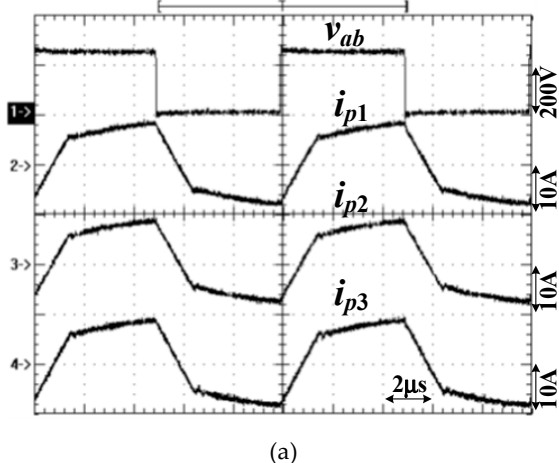

(a)

**Figure 6.** *Cont.*

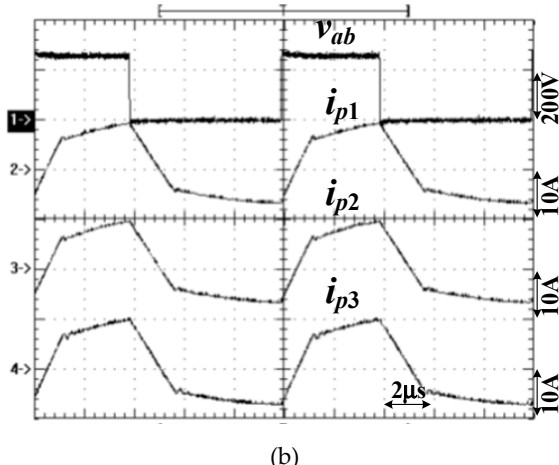

(b)

**Figure 6.** Measured three primary-side currents $i_{p1} \sim i_{p3}$ at rated power (**a**) with $V_{in}$ = 750 V and (**b**) with $V_{in}$ = 800 V.

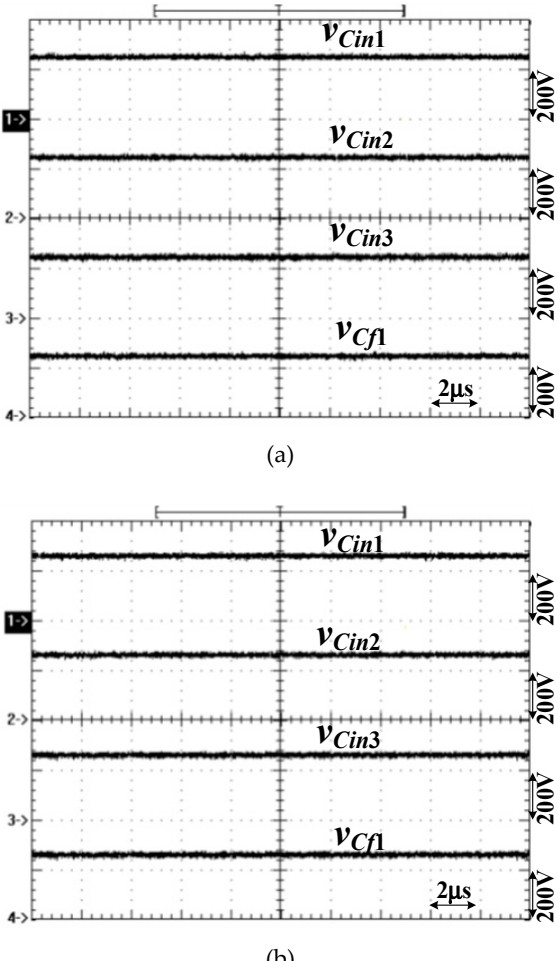

(a)

(b)

**Figure 7.** Measured results of the input split voltages and balance capacitor voltage at the rated power (**a**) with $V_{in}$ = 750 V and (**b**) with $V_{in}$ = 800 V.

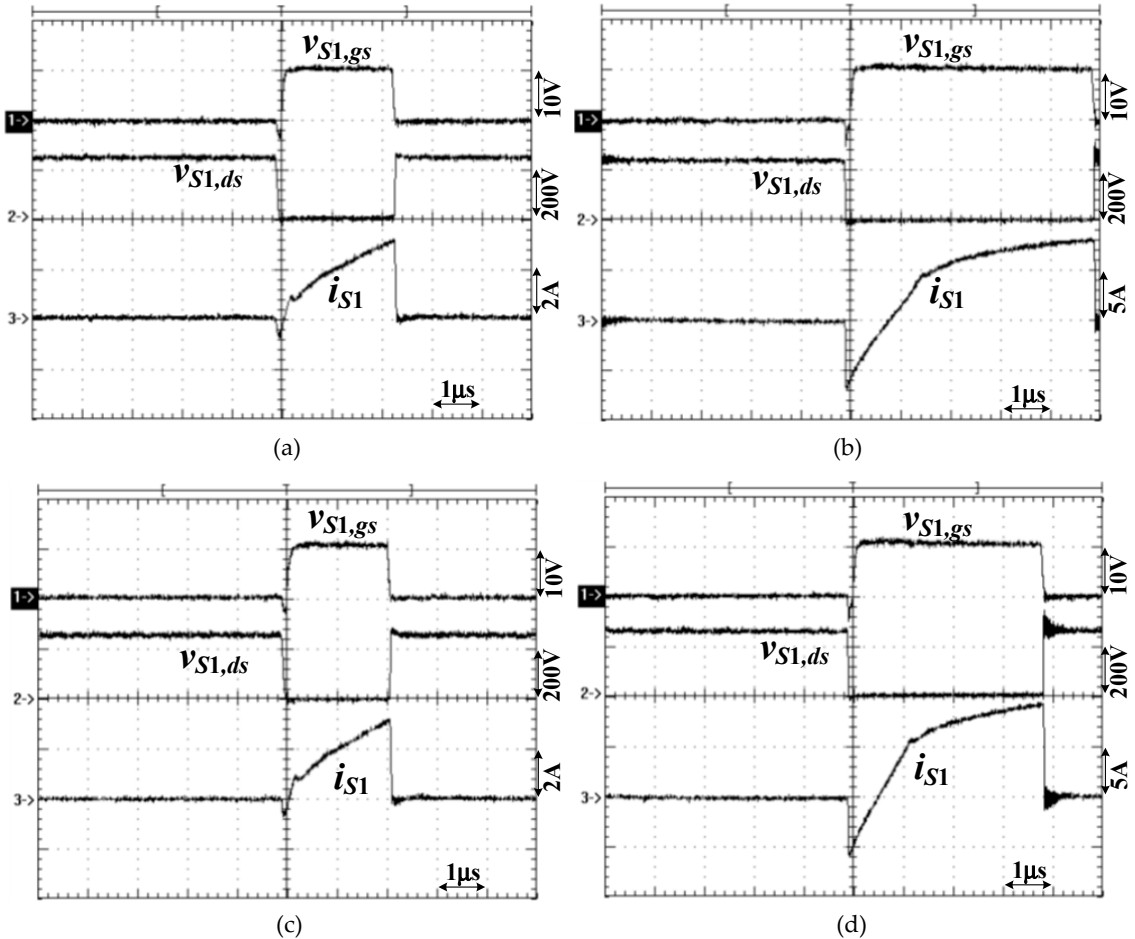

**Figure 8.** Measured results of the gating voltage and current of $S_1$ under (**a**) $V_{in}$ = 750 V and 20% output load, (**b**) $V_{in}$ = 750 V and the rated output load, (**c**) $V_{in}$ = 800 V and 20% output load, and (**d**) $V_{in}$ = 800 V and the rated output load.

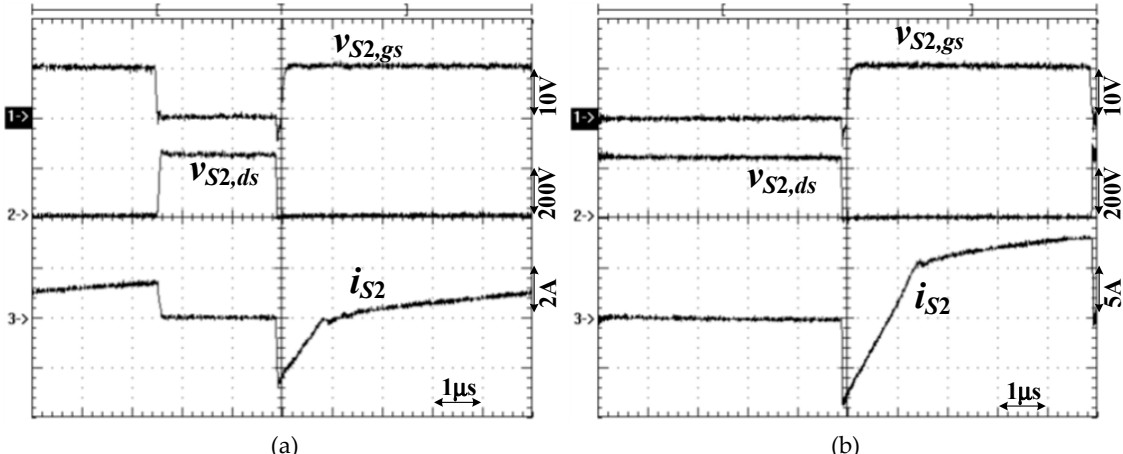

**Figure 9.** *Cont.*

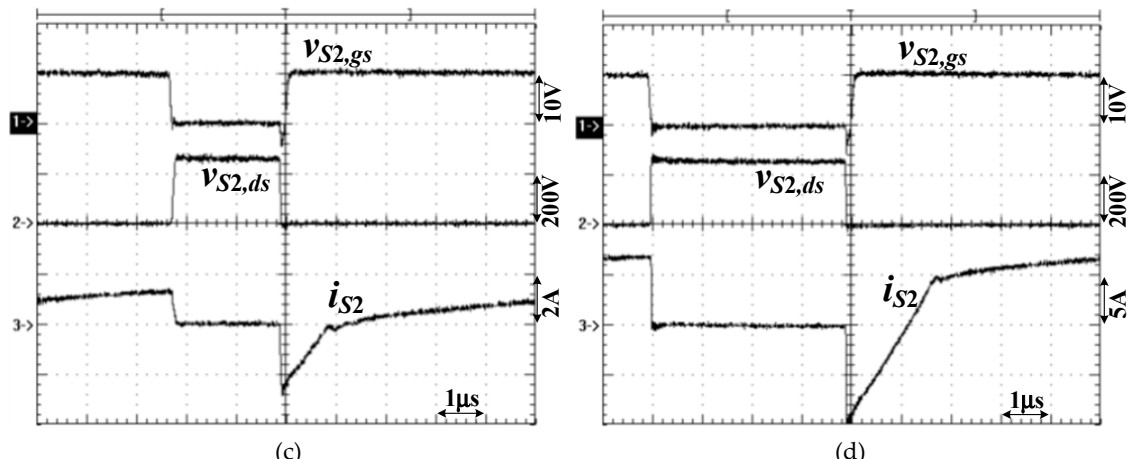

(c)　　　　　　　　　　　　　　　　(d)

**Figure 9.** Measured results of the gating voltage and current of $S_2$ under (**a**) $V_{in}$ = 750 V and 20% output load, (**b**) $V_{in}$ = 750 V and the rated output load, (**c**) $V_{in}$ = 800 V and 20% output load, and (**d**) $V_{in}$ = 800 V and the rated output load.

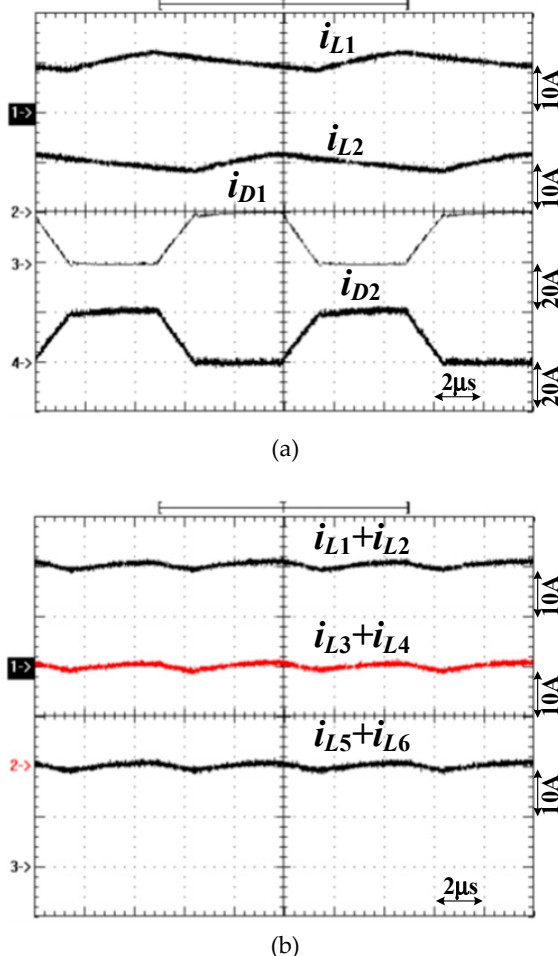

(a)

(b)

**Figure 10.** Measured waveforms of the secondary-side currents at the rated output power (**a**) $i_{L1}$, $i_{L2}$, $i_{D1}$, and $i_{D2}$ in first half bridge circuit and (**b**) output currents of three half bridge circuits.

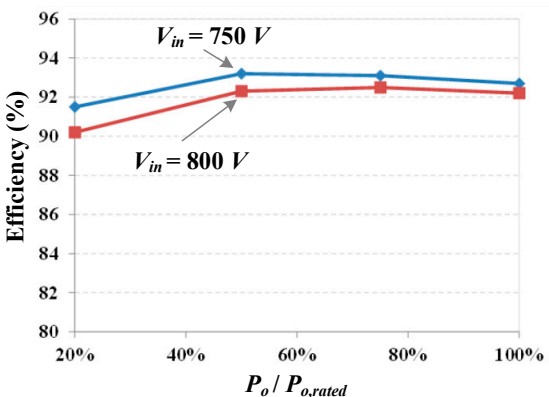

**Figure 11.** Measured efficiencies of the proposed converter.

## 5. Conclusions

A modular dc–dc converter with magnetic coupling current balance is presented for high power industry power units, dc light rail transportation, or dc microgrid system applications. Three half bridge circuits are employed in the proposed converter with primary-series secondary-parallel connection to lessen the voltage stress of power devices on high voltage side and current stress of passive components on low voltage side. Balance capacitors are used on high voltage side to achieve voltage balance on input split capacitors. In order to achieve current balance of three half bridge circuits, magnetic coupling current balance components are employed on the primary-side. The current doubler rectifiers are used on low voltage side to achieve partial ripple current reduction. Asymmetric pulse-width modulation approach is used to control power switches and regulate load voltage. From the experimental results, all power switches can be turned on at zero voltage from 20% output power. The other dc–dc converter topologies, such as full-bridge converter with phase-shift pulse-width modulation and resonant converter with frequency control, can also be applied in the proposed modular dc–dc converter with series-parallel connection with currents sharing and split voltages balance. Full-bridge circuit has two times of switch counts compared to the half-bridge circuits in the proposed converter. That will increase the cost and converter size. The resonant converter with frequency modulation cannot be designed at the optimal condition due to the switching frequency being related to the load condition and input voltage. Based on the analysis of circuit characteristics in Section 3 and the test results, it can be observed that the main drawbacks of the proposed converter, with asymmetric pulse-width modulation scheme, are unbalanced current rating on power devices $S_1{\sim}S_6$ and $D_1{\sim}D_6$, and inductors $L_1{\sim}L_6$, due to the duty cycle being related to the load current and input voltage. Similar, the average voltages on dc blocking capacitors $C_1{\sim}C_3$ are also related to duty cycle. If the proposed converter is operated under duty cycle equals 0.5, then the current rating of all power devices and inductors are balanced.

**Author Contributions:** B.-R.L. designed the main parts of the project and was also responsible for writing the paper.

**Funding:** This research was funded by Ministry of Science and Technology, Taiwan, under Grant MOST 105-2221-E-224-043-MY2.

**Acknowledgments:** The author would like to thank Yen-Chieh Huang for the help in the experimental results.

**Conflicts of Interest:** The author declares no potential conflict of interest.

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
