# Peer review of "Soft Switching DC Converter for Medium Voltage Applications"

_electronics, doi:10.3390/electronics7120449_

Round 1

Reviewer 1 Report

Here are my comments and questions with intent to help to improve the paper:

1.     The introduction can be improved with the advantages and disadvantages of other topologies employed for the same purpose.

2.     The assumption of same capacitances is taken into consideration (there are four different types of capacitances with several capacitor and, for each type of capacitance, all the corresponding capacitors are equal). Nevertheless, the capacitance values can undergo significant changes with regard to the rated value owing to the tolerances of the capacitors. What is the influence of these variations in the operation of the converter?

3.     The same question is applied to the inductances.

4.     What are the limits of the magnetic couplings?

5.     The results are satisfactory, but I miss a comparison with other topologies.

Author Response

Here are my comments and questions with intent to help to improve the paper:

1.     The introduction can be improved with the advantages and disadvantages of other topologies employed for the same purpose.

Answer: The discussion of the advantages and disadvantages of the proposed converter and the other topologies are discussed in the conclusion.

2.     The assumption of same capacitances is taken into consideration (there are four different types of capacitances with several capacitor and, for each type of capacitance, all the corresponding capacitors are equal). Nevertheless, the capacitance values can undergo significant changes with regard to the rated value owing to the tolerances of the capacitors. What is the influence of these variations in the operation of the converter?

Answer: Three kinds of capacitors: input split capacitors, voltage balance capacitors and dc voltage blocking capacitors are used in the proposed converter. Normally, the input split voltages are electrolytic capacitors and metallized polypropylene film (MPP) capacitors are used for voltage balance and dc voltage blocking. The voltages on input spilt capacitors and voltage balance capacitors are identical (vCin1=vCin2=vCin3=vCf1=vCf2) so that the conditions of capacitance variation on Cin1~Cin3 and Cf1~Cf2 are the same. Therefore, there is no problem for the capacitance variation on Cin1~Cin3 and Cf1~Cf2. The current sharing is used in the proposed circuit so that the primary side currents of three half-bridge circuits are identical. Thus, the capacitance variations on C1 ~ C3 are identical.

3.     The same question is applied to the inductances.

Answer: The current sharing with current balance magnetic-coupling component is used in the proposed circuit. Therefore, the primary-side and secondary-side currents of three half-bridge circuits are balanced.

4.     What are the limits of the magnetic couplings?

Answer: The adopted current balancing magnetic-coupling component is implemented by a magnetic core with high magnetizing inductance and unity turn-ratio. With high magnetizing inductance and unity turn-ratio, the primary-side and the secondary-side currents are balanced.

5.     The results are satisfactory, but I miss a comparison with other topologies.

Answer: The other dc-dc converter topologies such as full-bridge converter with phase-shift pulse-width modulation and resonant converter with frequency control can also be applied in the proposed modular dc-dc converter with series-parallel connection with currents sharing and split voltages balance. Full-bridge circuit has two times of switch counts compared to the half-bridge circuits in the proposed converter. That will increase the cost and converter size. The resonant converter with frequency modulation cannot be designed at the optimal condition due to the switching frequency is related to the load condition and input voltage. Based on the analysis of circuit characteristics in section 3 and the test results, it can observe that the main drawbacks of the proposed converter with asymmetric pulse-width modulation scheme are unbalance current rating on power devices S1 ~ S6 and D1 ~ D6 and inductors L1 ~ L6 due to the duty cycle is related to the load current and input voltage. Similar, the average voltages on dc blocking capacitors C1 ~ C3 are also related to duty cycle. If the proposed converter is operated under duty cycle equals 0.5, then the current rating of all power devices and inductors are balanced.

Reviewer 2 Report

This is a very well-written manuscript. However, I think some points can be made clearer: - It seems that the proposed topology can solve the voltage balancing problem of the multi-modular DC-DC converter. The modulation technique seems not so complicated. Does the topology have any drawbacks? If any, they should be pointed out in the introduction or conclusion sections. E.g. I think the design of magnetic components in the proposed topology may be troublesome? - In Figure 4 about the measured Gate-to-source voltages, the positive drive voltage is about 10V, that is reasonable for Si-MOSFET. However, the negative voltage spikes at turn on sometime reach -10V. Is this OK for the driver? What kind of driver was used for your experiment? - Although the author provide many experiment waveform, there is no efficiency or loss breakdown. I think such the analysis is necessary to evaluate the performance of the proposed topology.

Author Response

This is a very well-written manuscript. However, I think some points can be made clearer: - It seems that the proposed topology can solve the voltage balancing problem of the multi-modular DC-DC converter. The modulation technique seems not so complicated. Does the topology have any drawbacks? If any, they should be pointed out in the introduction or conclusion sections. E.g. I think the design of magnetic components in the proposed topology may be troublesome?

Answer: The limitation of the proposed converter is discussed in the conclusion. There is no troublesome for design the magnetic components.

- In Figure 4 about the measured Gate-to-source voltages, the positive drive voltage is about 10V, that is reasonable for Si-MOSFET. However, the negative voltage spikes at turn on sometime reach -10V. Is this OK for the driver? What kind of driver was used for your experiment?

Answer: I think there is a misunderstanding for Fig. 4. These four gate voltages are between 0V and 10V not -10V. The correction in Fig. 4 has been provided in this revision. The high-side MOSFET gate-driver with charge pump IC L6384 is used to drive power MOSFET.

- Although the author provide many experiment waveform, there is no efficiency or loss breakdown. I think such the analysis is necessary to evaluate the performance of the proposed topology.

Answer: The circuit efficiencies under different loads are provided and discussed in this revision.

Round 2

Reviewer 2 Report

The revised version is acceptable. I have no further comments.